# Fatty acid desaturases link cell metabolism pathways to promote proliferation of Epstein-Barr virus-infected B cells

Emmanuela N. Bonglack[1,2☉], Kaeden K. Hill[1☉], Ashley P. Barry[1], Alexandria Bartlett[1], Pol Castellano-Escuder[3], Matthew D. Hirschey[3], Micah A. Luftig [1]*

1 Department of Molecular Genetics and Microbiology, Center for Virology, Duke University School of Medicine, Durham, North Carolina, United States of America, 2 Cardiovascular Epidemiology Unit, University of Cambridge, Cambridge, United Kingdom, 3 Duke Molecular Physiology Institute, Duke University School of Medicine, Durham, North Carolina, United States of America

☉ These authors contributed equally to this work.
* micah.luftig@duke.edu

## Abstract

Epstein-Barr virus (EBV) is a gamma herpesvirus that infects up to 95% of the human population by adulthood, typically remaining latent in the host memory B cell pool. In immunocompromised individuals, EBV can drive the transformation and rapid proliferation of infected B cells, ultimately resulting in neoplasia. The same transformation process can be induced *in vitro*, with EBV-infected peripheral blood B cells forming immortalized lymphoblastoid cell lines (LCLs) within weeks. In this study, we found that the fatty acid desaturases stearoyl-CoA desaturase 1 (SCD1) and fatty acid desaturase 2 (FADS2) are upregulated by EBV and crucial for EBV-induced B cell proliferation. We show that pharmacological and genetic inhibition of both SCD1 and FADS2 results in a significantly greater reduction in proliferation and cell cycle arrest, compared to perturbing either enzyme individually. Additionally, we found that inhibiting either SCD1 or FADS2 alone hypersensitizes LCLs to palmitate-induced apoptosis. Further free fatty acid profiling and metabolic analysis of dual SCD1/FADS2-inhibited LCLs revealed an increase in free unsaturated fatty acids, a reduction of oxidative phosphorylation, and a reduction of glycolysis, thereby linking the activity of SCD1 and FADS2 to overall growth-promoting metabolism. Lastly, we show that SCD1 and FADS2 are important in the growth of clinically derived EBV+ immunoblastic lymphoma cells. Collectively, these data demonstrate a previously uncharacterized role of lipid desaturation in EBV+ transformed B cell proliferation, revealing a metabolic pathway that can be targeted in future anti-lymphoma therapies.

**Data availability statement:** All relevant data are within the manuscript and its Supporting information files.

**Funding:** M.A.L. received support from the National Cancer Institute (R01-CA140337). E.N.B. was supported by F31-CA257621. The funders had no role in study design, data collection and analysis, decision to publish, or preparation of the manuscript.

**Competing interests:** The authors have declared that no competing interests exist.

## Author summary

Epstein-Barr virus (EBV) is a ubiquitous herpesvirus that latently infects nearly all adults worldwide. However, EBV-associated disease is prevented in most due to continuous and robust immune control. In the absence of such control, EBV oncoproteins can promote uncontrolled cell proliferation nudging those cells towards cancer. In particular, the viral proteins are sufficient to rewire cell metabolism, thereby driving rapid proliferation of its major latent infection target, B cells. Previous work has shown that EBV induces nucleotide metabolism, high levels of aerobic glycolysis, and fatty acid biosynthesis in B cells. In this study, we show that fatty acid desaturation is another important part of this EBV-driven metabolic rewiring that allows infected B cells to proliferate. We show that EBV drives the upregulation of SCD1 and FADS2: two desaturases acting in parallel pathways that can compensate for each other to promote cell division and oxidative metabolism, while preventing lipotoxicity. Our data can help in our understanding of how EBV promotes cancer, hopefully aiding in the development of metabolism-based treatment strategies for viral lymphomas.

## Introduction

Epstein-Barr virus infects most individuals worldwide by adulthood. While primary infection is often asymptomatic, it can manifest as infectious mononucleosis in adolescents and is followed by a lifelong latency period in memory B cells [1,2]. *In vivo*, EBV viral propagation is kept in check by the host immune system, with up to 1–2% of circulating CD8+T cells in asymptomatic individuals being specific for EBV epitopes [3]. In the absence of an intact immune system, such as following an organ transplant or in HIV/AIDS infection, EBV reactivation can lead to several lymphomas and lymphoproliferative disorders, such as post-transplant lymphoproliferative disease (PTLD) [4]. Additionally, more than a third of AIDS-related non-Hodgkin's lymphomas, such as Burkitt Lymphoma, Diffuse Large B Cell Lymphoma (DLBCL), and Primary Effusion Lymphoma (PEL), are EBV-positive, highlighting the far-reaching oncogenic potential of EBV in contexts of immune-suppression [5].

*In vitro*, EBV infection of primary B cells from peripheral blood gives rise to a transient period of hyperproliferation, followed by a DNA damage-driven growth arrest in >95% of infected cells [6]. However, a small subset of these cells evade host proliferation barriers like the DNA Damage Response (DDR) pathway, ultimately becoming indefinitely proliferating lymphoblastoid cell lines (LCLs) within approximately five weeks. Moreover, viral and host gene expression profiles following primary B cell infection mimic those observed in EBV-positive lymphomas [7,8]. Thus, LCLs serve as a model for studying molecular drivers of EBV-related lymphomagenesis.

During EBV-mediated B-cell immortalization *in vitro*, host responses like the DDR and metabolic stress play a significant cell-intrinsic role in preventing EBV-driven B-cell transformation [6,9,10]. While metabolic pathways like one-carbon metabolism,

autophagy, lactate transport, and cholesterol metabolism have been studied in detail, more remains to be understood about fatty acid desaturation [6,9,11–13]. Fatty acid desaturation is the process of introducing a double bond to a lipid substrate, which promotes the synthesis of diverse mono- and polyunsaturated fatty acids (MUFAs and PUFAs) in cells. These resulting MUFAs and PUFAs can then be used for a range of cellular processes, including membrane biosynthesis and remodeling, energy storage via triacylglycerols, cholesterol esters, and lipid droplets, as well as cellular signaling in the context of immunity [14]. Fatty acid desaturation is mediated by desaturase enzymes, which have substrate and location (carbon) specificities intricately tied to their expression patterns and functions [14,15]. As such, fatty acid desaturation and associated elongation are critical for lipid diversity in cells.

Fatty acid desaturation in humans is primarily driven by three desaturases: Stearoyl-CoA Desaturase (SCD1) and Fatty Acid Desaturase 1 and 2 (FADS1 and FADS2) [16]. SCD1 is transcriptionally activated by sterol regulatory binding protein 1 (SREBP1), a key regulator of *de novo* lipid synthesis that also promotes expression of FADS1 and FADS2, the latter of which is additionally regulated by the peroxisome proliferator-activated receptor-alpha (PPAR-α) [17–19]. SCD1 is ubiquitously expressed across tissue types, but its function is limited to the desaturation of saturated fatty acids (SFAs). As such, its two main substrates are palmitate and stearate, yielding the MUFAs palmitoleate and oleate, respectively [15].

Similar to SCD1 in its ubiquitous nature, FADS2 is widely expressed in humans, but its substrate profile includes at least six lipid species. These fatty acid substrates include but are not limited to linoleic acid (LA), α-linoleic acid, docosapentaenoic acid (DPA), and eicosadienoic acid [20,21]. FADS1 catalyzes the synthesis of arachidonic acid (AA) from dihomo-γ-linolenic acid (DGLA), and eicosapentaenoic acid (EPA) from eicosatetraenoic acid (ETA) [20]. Both AA and EPA are involved in lipid signaling, with AA being an inflammatory intermediate and EPA driving macrophage activation [22,23]. The FADS2-mediated desaturation of LA, DPA, and eicosadienoic acid yields γ-linolenic acid (GLA), docosahexaenoic acid (DHA), and dihomo-γ-linolenic acid (DGLA), respectively. Because of the complex role of PUFAs in cell membrane dynamics and cell signaling, FADS1 and FADS2 activity are implicated in a variety of inflammatory conditions and metabolic diseases [24–26].

Given their well-documented role in maintaining cell homeostasis, SCD1 and FADS2 have been implicated in tumorigenesis. Reduced SCD1 expression and activity have been associated with a significantly reduced risk of developing breast cancer, while higher SCD1 expression has been associated with worsened clinical outcomes in breast cancer [27]. Additionally, elevated SCD1 expression and, subsequently, palmitoleate levels have been linked with neoplastic transformation in prostate, colon, kidney, and lung cancer [28]. Of all three human desaturases, FADS2 and SCD1 are most frequently dysregulated in cancer, with FADS2 overexpression being associated with increased disease progression in uveal melanoma, bladder urothelial carcinoma, clear cell renal carcinoma, and lung and pancreatic adenocarcinomas [29]. FADS2 has also been found to be overexpressed in breast cancer [30]. In recent studies, FADS2 acted noncanonically to convert the SCD1 substrate palmitate to sapienate in SCD1-independent cancer cells [31,32], highlighting the highly plastic nature of FADS2 and identifying a critical interplay between SCD1 and FADS2 in tumor biology.

While the role of fatty acid desaturases has not yet been explored in EBV-infected B cells, the viral protein Latent Membrane Protein 1 (LMP1) has been shown to upregulate SREBP1 in EBV-positive nasopharyngeal carcinoma (NPC) cell lines [33]. Furthermore, the EBV Nuclear Antigen 2 (EBNA2) occupies the SREBP2 promoter as part of a larger effort by the virus to hijack fatty acid pathways and promote virus-driven proliferation and survival in B lymphocytes [11], and activation of lymphocytes has been associated with an increase in PUFAs mediated by FADS2 [34,35]. We sought to address the role and regulation of SCD1 and FADS2 during EBV-driven tumorigenesis, particularly considering the role of EBV as a mitogen. We assess how the expression and regulation of SCD1 and FADS2 during EBV-driven immortalization impacts B cell growth, survival, and, more broadly, cellular lipid metabolism. Our findings highlight previously unidentified metabolic dependencies in EBV-driven B cell immortalization and provide a rationale for further exploring desaturase inhibitors as therapeutic targets in EBV-associated lymphomas and lymphoproliferative disorders.

## Methods

### Cell culture conditions and viruses

Buffy coats were obtained from normal human donors through the Gulf Coast Regional Blood Center (Houston, TX), followed by peripheral blood mononuclear cells (PBMCs) isolation by a Ficoll Histopaque-1077 gradient (Sigma, H8889). B95-8 strain of Epstein-Barr virus was produced from the B95-8 Z-HT cell line as previously described [36]. Virus infections were performed in bulk by adding 50 μL of filtered B95-8 supernatant to $1\times10^6$ PBMCs.

Early EBV-infected PBMCs and uninfected B cells were kept in RPMI 1640 medium supplemented with 15% heat-inactivated fetal bovine serum (Corning), 2 mM L-Glutamine, 100 μg/ml penicillin with 100 μg/ml streptomycin (ThermoFisher), and 0.5 μg/mL Cyclosporine A (Sigma). Uninfected PBMCs were stimulated using either 2.5 μg/mL CpG, or 5 ng/mL rhCD40L (R&D Systems, Cat# 6420-CLB)+200ng/mL αHA crosslinker (R&D Systems, Cat# MAB060) and 20 ng/mL IL-4 (Pepro Tech, Cat# 200-04-20UG). All LCLs were kept in RPMI 1640 medium supplemented with 10% heat-inactivated fetal bovine serum (Corning). LCLs that were transfected with a gene-targeting Cas9-RNP complex for CRISPR knockout were cultured in RPMI 1640 medium supplemented with 15% heat-inactivated fetal bovine serum (Corning). IBL-1 cells were also cultured in RPMI 1640 medium supplemented with 10% heat-inactivated fetal bovine serum (Corning).

### Gene expression analyses

RNA sequencing or DNA microarray data presented in this study were from previously published datasets [6,13,37]. Total RNA for qPCR was isolated from LCLs treated with either DMSO or SCD1i+FADS2i (10 μM A939572 + 20 μM SC-26196) using the Qiagen RNeasy Mini Kit (Qiagen, Cat# 74104). Individual LCL donors were treated as biological replicates. qPCR was conducted using SYBR Green (Quanta Biosciences) and an Applied Biosystems QuantStudio 6 Pro real time qPCR system (ThermoFisher). Primer sets used were EBNA1 (forward: 5' TGCCTGAACCTGTGGTTGG 3', reverse: 5' CATGATTCACACTTAAAGGAGACGG 3'), EBNA2 (forward: 5' GCTTAGCCAGTAACCCAGCACT 3', reverse: 5' TGCTTAGAAGGTTGTTGGCATG 3'), EBNA3A (forward: 5' GCTTAGCCAGGTAACTTAGG 3', reverse: 5' GCCTGTCCTTGTCCATTTTG 3'), LMP1 (forward: 5' AATTTGCACGGACAGGCATT 3', reverse: 5' AAGGCCAAAAGCTGCCAGAT 3'), LMP2A (forward: 5' CGGGATGACTCATCTCAACACATA 3', reverse: 5' GGCGGTCACAACGGTACTAACT 3'), BZLF1 (forward: 5' ACGACGCACACGGAAACC 3', reverse: 5' CTTGGCCCGGCATTTTCT 3'), BMRF1 (forward: 5' CGTGCCAATCTTGAGGTTTT 3', reverse: 5' CGGAGGCGTGGTTAAATAAA 3'), SETDB1 (forward: 5' GACTACAATACCGGGACAGTAGC 3', reverse: 5' CCCAGCATCACCTGAATCAAT 3').

### Gene set enrichment analysis

Gene set enrichment analysis (GSEA) was run on DNA microarray data that was previously published by our lab [6] using GSEA v4.3.2 [38]. The heatmap of top 20 differentially expressed genes (DEGs) was generated by sorting the entire list of significant DEGs by fold change from uninfected B cell to LCL. The upregulation of fatty acid desaturation from B cell to LCL was assessed using the KEGG Biosynthesis of Unsaturated Fatty Acids gene set.

### CRISPR-Cas9 gene editing

To achieve *scd1, fads2, mdm2,* or *cd46* gene knockout, LCLs were transfected with a ribonucleoprotein (RNP) complex containing Cas9 protein duplexed with gene-targeting single-guide RNAs (sgRNAs). To assemble the complexes, TrueCut Cas9 protein v2 (ThermoFisher, Cat# A36498) was incubated with sgRNAs (3 sgRNAs per target gene) (Synthego) at a ratio of 1:3 (Cas9 protein: total sgRNA). Next, the assembled Cas9-RNP complex was transfected into LCLs via electroporation, using the ThermoFisher Neon Transfection System. The non-essential surface protein, CD46, was used as a marker of successful transfection, as described previously [39]. Therefore, all cells were transfected with CD46 targeting RNP complexes in addition to RNP complexes targeting the gene of interest. The absence of CD46 as determined by

FACS, was used as a proxy to identify cells with the gene(s) of interest knocked out. The following sgRNAs were used: *SCD1* (sgRNA1–5'GCCACCGCUCUUACAAAGCU 3', sgRNA2–5' UACCUGGAAUGCCAUUGUGU 3', sgRNA3–5' AGGAGGGAUCAGCACCAGAG 3'); *FADS2* (sgRNA1–5'CACCGACACCUCGCGCUCGG 3', sgRNA2–5'CCAGCCAC-CUGUCGGUGCGC 3', sgRNA3–5'CCGAUGACCCGCUGGCCCCC 3'); CD46 (sgRNA1–5'GAGAAACAUGUCCAUAU-AUA3', sgRNA2–5'AACUCGUAAGUCCCAUUUGC3', sgRNA3–5'UUGCUCCUUAGAGGAAAUA3').

Because targeting *fads2* with Cas9-RNP complexes did not fully eliminate FADS2 protein abundance as it did SCD1, we validated *fads2* gene editing by isolating genomic DNA from LCLs sorted based on CD46 negativity using genomic DNA isolation kits (PureLink Genomic DNA Mini Kit, ThermoFisher). We then sequenced *fads2* in the targeted (CD46-) and untargeted (CD46+) populations and calculated the indel percentage using Synthego's Inference of CRISPR Editing tool (Synthego Performance Analysis, ICE Analysis. 2019. v3.0. Synthego; 2025). Primers used to amplify genomic *fads2* over a region spanning the sgRNA cutsites were: forward: 5' ACTGGAGGCAAAAGTCCATAGC 3', reverse: 5' GGGAC-GAGCTTTCCTTCTCG 3'.

### Flow cytometry and sorting

To track proliferation in early EBV-infected B cells, cells were stained with the fluorescent proliferation tracking dye Cell-Trace Violet (ThermoFisher, Cat# C34557). For subsequent surface antibody staining, cells were washed in FACS buffer (2% FBS in PBS), followed by incubation with the appropriate antibody for 20 min at 4°C in the dark, and finally, washing once more with FACS buffer before analysis on a BD FACS Canto II cytometer. B cells were purified from peripheral blood mononuclear cells (PBMCs) using the BD Human B Lymphocyte Enrichment Set (BD Biosciences Cat# 558007). To track CD46-positivity in Cas9-targeted LCLs, cells were stained with the fluorescent APC mouse anti-human CD46 antibody (Biolegend Cat# 352405). Briefly, cells were washed in cold FACS buffer, followed by incubation with the antibody at a dilution of 1 µL/million cells for 20 min at 4°C in the dark. This was followed by an additional wash with FACS buffer and subsequent analysis on a BD FACS Canto II cytometer. To sort CD46-negative/positive populations for Western blot vali-dation or genomic DNA sequencing, LCLs transfected with the targeting Cas9-RNP complexes were sorted using a Sony SH800 Cell Sorter at the Duke Cancer Institute Flow Cytometry Core.

### Cell growth assays

LCLs or IBL-1 cells were seeded at a density of $2.5 \times 10^5$ cells/mL and treated with either dimethyl sulfoxide (DMSO), A939572 (Sigma, Cat# SML2356), or SC-26196 (Sigma, Cat# PZ0176). DMSO concentrations were kept constant between treatments and didn't exceed 0.5% DMSO. When exogenous fatty acids were supplemented, BSA-conjugated palmitate (Cayman Chemicals, Cat# 29558), BSA-conjugated oleate (Cayman Chemicals, Cat# 29557), or BSA control (Cayman Chemicals, Cat# 29556) were used. The concentration of BSA was kept constant between treatments. Cell growth was assessed daily using either CellTiter Glo (Promega) with a Glo Max Explorer plate reader (Promega) or Try-pan exclusion with an automated cell counter (Countess II, ThermoFisher).

### Cell death assays

Cell viability was assessed using Trypan exclusion or the DNA-binding fluorescent dye, CellTox Green (Promega). Apoptosis was measured using Caspase 3/7 Glo (Promega) with a Glo Max Explorer plate reader (Promega), Annexin V staining and flow cytometry, or cleaved caspase 3/7 staining and flow cytometry. For flow cytometry-based methods, cells were first washed in FACS buffer before being stained with CellEvent Caspase 3/7 Green (ThermoFisher) for 30 min at 37°C and 5% CO2. Then, cells were diluted in 1X Annexin V binding buffer (eBioscience) before staining with Pacific Blue-conjugated Annexin V (BioLegend, Cat# 640918) and 7-AAD (BioLegend, Cat# 420404) for 15 min at room tempera-ture in the dark. Samples were then further diluted in 1X Annexin V Binding Buffer before analysis on a BD FACS Canto II Cytometer. Before data analysis, standard fluorophore compensation techniques were applied using FlowJo.

## Immunoblotting

Cells were pelleted and washed in PBS, then lysed in RIPA Lysis Buffer (ThermoFisher Cat# 89900) with complete protease and phosphatase inhibitors. All protein lysates were run on NuPAGE 4–12% gradient gels (ThermoFisher) and transferred to a PVDF membrane (BioRad). Membranes were blocked in 5% milk in TBST and stained with primary antibody overnight at 4°C, followed by a wash and staining with secondary HRP-conjugated antibody for 1 hour at room temperature. Membranes were imaged using a LI-COR Odyssey XF Imager. Antibodies used include SCD1 (ThermoFisher, Cat# PA595762; 1:500 dilution), FADS2 (ThermoFisher, Cat# PA5–87765; 1:500 dilution), and MAGOH (Santa Cruz Biotechnology, Cat# sc-56724; 1:1000 dilution). MAGOH was used as a loading control because its expression does not change during EBV-mediated B cell outgrowth.

## BrdU pulse for cell cycle analysis

After either 48 or 72 hours of the indicated treatment, 10 μM BrdU (BD Biosciences, Cat# 552598) was added to the culture medium and incubated at 37°C and 5% $CO_2$ for two hours. Then, cells were harvested and washed with FACS buffer before fixation, permeabilization, and DNase treatment to reveal incorporated BrdU. Cells were stained with a FITC-conjugated BrdU antibody (BioLegend, Cat# 364104) for 20 min at room temperature in the dark, before staining of total DNA with 7-AAD (BioLegend, Cat# 420404) for 15 min. BrdU and 7-AAD fluorescence were then analyzed on a BD FACS Canto II cytometer.

## Mass spectrometry-based fatty acid profiling

We collaborated with the Mass Spectrometry Core Laboratory at the University of North Carolina-Chapel Hill to measure the abundance of 31 free fatty acid species in dual SCD1/FADS2-inhibited LCLs. After either 48 or 72 hours of treatment with 10 μM SCD1i+20 μM FADS2i, cells were harvested, washed with ice cold PBS, and flash frozen. Fatty acid methyl esterification (FAME) was conducted using methyl tert-butyl ether to efficiently separate free fatty acids, before extraction in hexane. Methyl-esterified free fatty acids were analyzed using a ThermoFisher Exactive GC mass spectrometer (ThermoFisher), and ions were introduced by electron impact ionization (70eV). The mass range was set to 50–400 *m/z*. All measurements were recorded at a resolution setting of 60,000. Separations were conducted on an Agilent J&W HP-88 column (100 m x 0.25 mm ID x 0.20 μm film). GC conditions were set at 100°C to start and maintained for 13 min. The system was then heated at 10°C/min up to 180°C, held for 6 min, then brought to 192°C at 1°C/min, held for 9 min, then heated to 230°C at 10°C/min and held for 10 min (total method 63min) Injection volume for all samples was 1μL with a splitless injection. The carrier gas (helium) was held at constant flow of 1.3mL/min. Xcalibur (ThermoFisher) was used to analyze the data. Molecular formula assignments were determined with Molecular Formula Calculator (v 1.2.3). All observed species were singly charged, as verified by unit *m/z* separation between mass spectral peaks corresponding to the $^{12}C$ and $^{13}C^{12}C_{c-1}$ isotope for each elemental composition.

## Seahorse analysis

ECAR, OCR, and rate of ATP synthesis were measured using the Seahorse XFe96 Real-Time ATP Rate assay (Agilent). LCLs were adhered to cell culture plates using poly-D-lysine and Dulbecco's PBS (Gibco). ECAR and OCR were measured in XF RPMI Base Medium (Agilent) supplemented with 1mM sodium pyruvate, 2 mM L-glutamine, and 10 mM glucose (Sigma-Aldrich). Supplemented LCLs were seeded into wells at 250,000 cells per well, centrifuged briefly so the cells form a monolayer, and placed in a non-$CO_2$ incubator for one hour before analysis. OCR and ECAR were measured over time, with injections of oligomycin and rotenone+antimycin A.

## Statistical methods and analysis of fatty acid profiling results

All statistical analyses were conducted using GraphPad Prism v10.2.3. All unpaired statistical comparisons were made using a two-tailed, unpaired Student's T test or an ANOVA with Tukey's post-hoc test. All paired statistical comparisons

were made using a two-tailed, paired Student's T test. A confidence level of 0.05 was used for all tests. Free fatty acid profiling data were analyzed using a custom R script, where data were first imputed using the Pomalimpute functions, before normalization using the PomaNorm function. Because correlation analysis revealed high levels of colinear trends between fatty acid species, differential abundance analysis was conducted using limma. Treatment and treatment time were considered fixed effects, while donor was treated as a random effect.

## Results

### EBV infection of B lymphocytes leads to upregulation of lipid desaturases

The fatty acid desaturase enzymes stearoyl Co-A desaturase (SCD1) and fatty acid desaturase 2 (FADS2) are two of the twenty most highly upregulated genes following B cell immortalization with EBV infection as evidenced from prior microarray studies [6] (Fig 1A). Consistent with the role of SCD1 and FADS2 in the biosynthesis of unsaturated fatty acids, we found that the Kyoto Encyclopedia of Genes and Genomes (KEGG) set of genes involved in the biosynthesis of unsaturated fatty acids was enriched in LCLs relative to uninfected B cells (Fig 1B). Furthermore, previously published temporal transcriptomic and proteomic data confirm increased levels of SCD1 and FADS2 throughout EBV-infected B-cell outgrowth (Fig 1C, D) [11,40]. We confirmed these findings by Western blot, observing a time-dependent increase in SCD1 and FADS2 expression post-infection, compared with primary uninfected B cells (Figs 1E and S1).

During EBV infection, EBV Nuclear Antigen 2 (EBNA2) acts as a master transcriptional regulator, recruiting various transcription factors shortly after infection to promote the expression of pro-growth genes, including several critical metabolic enzymes and factors like the one-carbon metabolism enzyme MTHFD2 and the lactate transporter MCT1 [13,41]. Given its documented role in driving changes in host metabolism and the early expression dynamics of SCD1 and FADS2, we assessed whether EBNA2 might drive SCD1/FADS2 expression using the estrogen-inducible P493-6 system [42]. Upon inducing EBNA2 expression by adding β-estradiol, we observed a subsequent increase in both SCD1 and FADS2 RNA levels along with c-MYC and MTHFD2 as controls (Fig 1F). These data confirm an EBNA2-mediated mode of SCD1 and FADS2 upregulation within the context of EBV infection.

### Pharmacological inhibition of SCD1 and FADS2 suppresses growth of EBV-infected and stimulated primary B cells

Given the high expression levels of SCD1 and FADS2 in LCLs, we wondered whether they might be critical for EBV-driven LCL growth. To assess this, we used the small molecule inhibitors A939572 (SCD1i) and SC-26196 (FADS2i) and observed that inhibition of either SCD1 or FADS2 in LCLs led to a dose-dependent reduction in cell growth by 48 hours post treatment (Figs 2A–D, S2A and S2B). While individual SCD1 and FADS2 inhibition yielded significant decreases in LCL growth, their collective role as desaturase enzymes begged the question of whether they might be acting in a complementary manner to create diverse desaturated lipid species that favor growth, as demonstrated in breast cancer cells [31]. Thus, we assessed the dual inhibition of SCD1 and FADS2 and found that this led to a marked reduction in cell growth over time compared to treatment with SCD1i or FADS2i individually (Fig 2E). To formally assess whether the drugs were acting in an additive or synergistic manner, we generated isobolograms [43] using the IC50 for SCD1i and a sub-IC50 value for FADS2i (due to solubility constraints). Treatment with as low as 2.5 μM FADS2i reduced the IC50 of SCD1i more than twofold (S2C and S2D Fig). Additionally, the combination index at 2.5 μM FADS2i using the sub-IC50 value of 50 μM is 0.44, indicating a synergistic effect. This value would only decrease if the IC50 of FADS2i were used. Therefore, the effect of the drug combination is more than the sum of each drug's effect individually.

This decrease in LCL growth upon combined SCD1 and FADS2 inhibition could have been due to either cell death or cell cycle arrest. To tease this apart, we assessed the extent to which cell cycle progression, viability, and apoptosis were impacted by desaturase inhibition. We stained desaturase inhibitor-treated LCLs with BrdU and 7-AAD and measured cell cycle progression using flow cytometry. After 48 hours of drug treatment, we observed that single SCD1-inhibited and

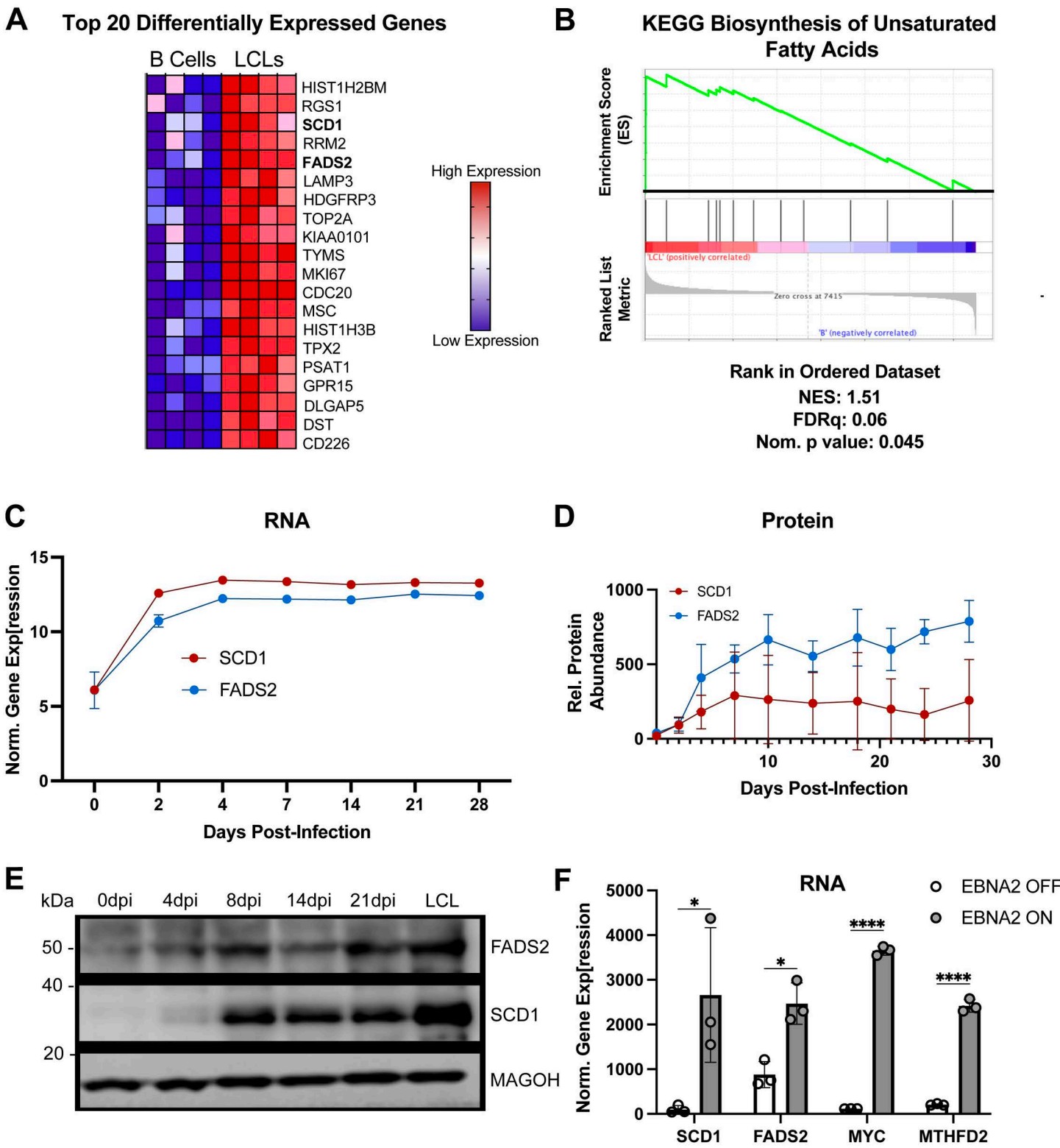

**Fig 1. EBV infection upregulates SCD1 and FADS2 in B cells.** (A) Microarray heatmap of the top 20 differentially expressed genes between B cells and LCLs. (B) Gene set enrichment between B cells and LCLs for the KEGG Biosynthesis of Unsaturated Fatty Acids pathway, using microarray data.

Normalized enrichment score (NES), false discovery rate (FDRq), and nominal p value are shown below the plot. (C) Bulk RNA-seq data [37] showing SCD1 and FADS2 mRNA levels throughout early EBV-infection. Y-axis units are reads per kilobase million (RPKM). (D) SCD1 and FADS2 protein levels throughout EBV-driven B cell outgrowth measured previously via mass spectrometry [40]. (E) SCD1 and FADS2 protein levels at days 0, 4, 8, 14, 21, and 6-weeks post EBV infection (LCL). MAGOH was used as a loading control because its expression is constant during EBV-driven B cell outgrowth. (F) RNA-seq data [13] from P493-6 cells that express EBNA2 fused to a modified estrogen receptor, allowing EBNA2 to be activated post-translationally by the administration of β-estradiol ("EBNA2-ON"). MYC and MTHFD2 expression levels were included as positive controls for EBNA2 activity. Y-axis units are fragments per kilobase million (FPKM) (n = 3, from 3 donors). Statistical significance was determined using a Student's unpaired T-test. For all pairwise comparisons: * p < 0.05, **p < 0.005, ***p < 0.0005, ****p < 0.00005.

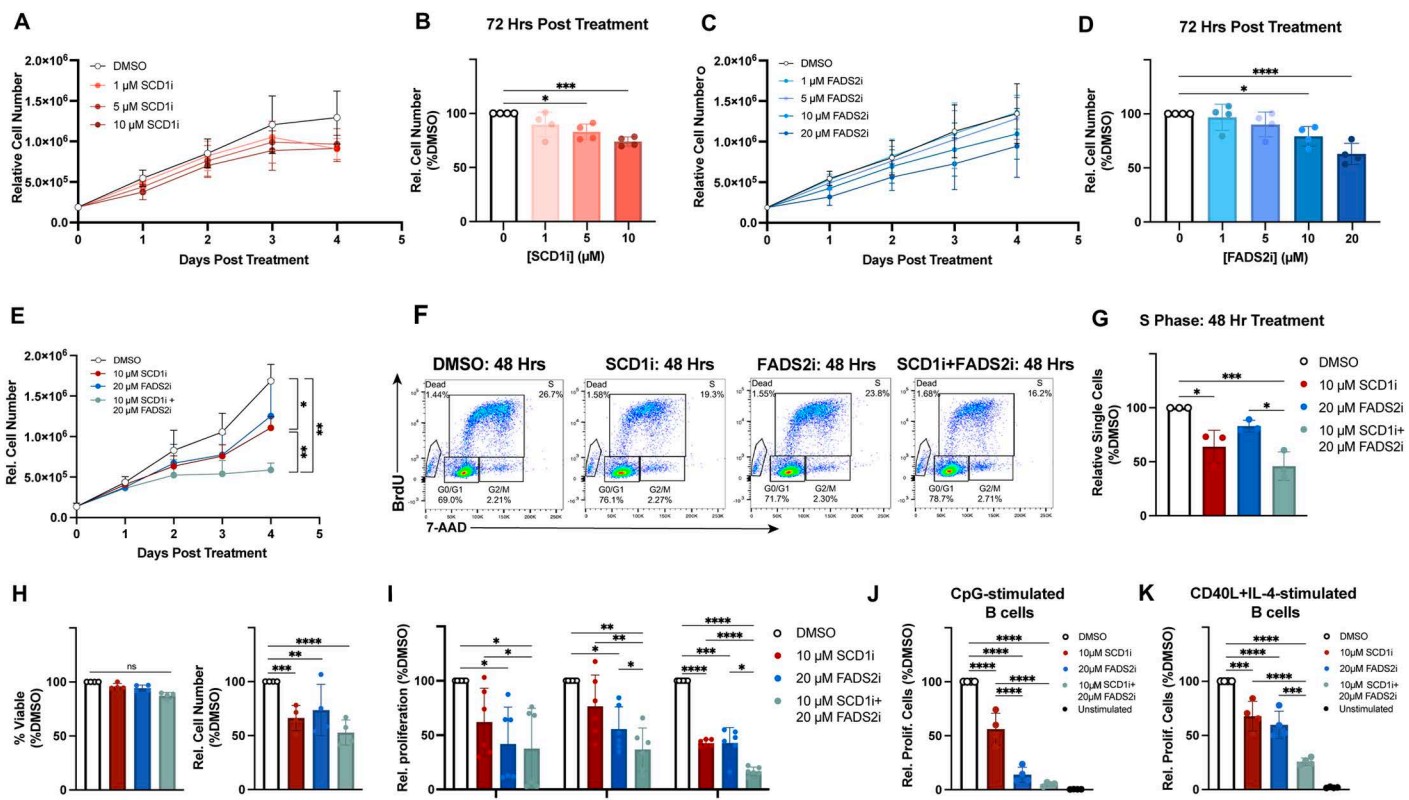

**Fig 2. Inhibition of SCD1 and FADS2 reduces B cell outgrowth.** (A-B) LCL growth in the presence of indicated doses of A939572 (SCD1i) (n = 4, LCLs from 4 donors). Growth was measured using Cell Titer Glo. (C-D) LCL growth in the presence of SC-26196 (FADS2i) (n = 4, LCLs from 4 donors). Growth was measured as in A-B. (E) LCL growth in the presence of 10 μM SCD1i, 20 μM SC-26196 FADS2i, or both inhibitors (n = 4, LCLs from 4 donors). Growth was measured as in A-B. (F) Representative dot plots, gated on live and dead single cells, showing anti-BrdU (FITC) versus total DNA (7-AAD) to measure the number of cells in each stage of the cell cycle at 48 hours post treatment. (G) DMSO-normalized relative percentage of single cells in S phase after 48 hours of drug treatment (n = 3, LCLs from 3 donors). (H) DMSO-normalized relative live cell percentage and number after 72 hours of treatment with indicated drugs (n = 4, from 4 donors). Cell number measured using Trypan dye. (I) Bulk PBMCs (n = 6, PBMCs from 2 donors) were treated with 10 μM SCD1i and/or 20 μM FADS2i immediately following EBV infection, and CD19 + B cell proliferation was based on CellTrace Violet dilution over time. (J) PBMCs (n = 4, PBMCs from 2 donors) were stimulated with 2.5 μg/mL CpG and treated with 10 μM SCD1i and/or 20 μM FADS2i. CD19 + B cell proliferation was assessed based on Cell Trace Violet dilution after five days. (K) PBMCs (n = 4, PBMCs from 2 donors) were stimulated with 5 ng/mL CD40L+20 ng/mL IL-4 and treated with 10 μM SCD1i and/or 20 μM FADS2i. CD19 + B cell proliferation was assessed based on Cell Trace Violet dilution after five days. Statistical significance for pairwise comparisons determined using a Tukey's post-hoc test or Dunnett's post-hoc test (panels B & D) (* p < 0.05, **p < 0.005, ***p < 0.0005, ****p < 0.00005).

single FADS2-inhibited LCLs displayed a significantly lower percentage of cells in S phase, further exacerbated upon dual inhibition of both SCD1 and FADS2 (Fig 2F–G). We did not observe an increase in cell death after 72 hours of treatment with the SCD1 and FADS2 inhibitors, confirming that growth reduction in LCLs following SCD1 and FADS2 inhibition is primarily due to a G1/S-phase growth arrest, not cell death (Fig 2H).

As EBV latent infection proceeds through a series of cell fate and activation transitions early after B cell infection before LCL outgrowth, we assessed the role of SCD1 and FADS2 in the early events in B cell transformation. Despite significant B cell donor variation, we observed a modest reduction in B cell growth following single SCD1 or FADS2 inhibition on days 4 and 7 post-infection. However, this single inhibitor-induced growth reduction phenotype was more robust on day 10-post infection, suggesting a more substantial contribution towards growth in slightly later (>1 week) stages of outgrowth compared to very early (<1 week) stages (Fig 2I). Like LCLs, dual SCD1 and FADS2 inhibition was detrimental to early-infected B cell proliferation, more so than either desaturase inhibited alone. Because EBV lytic reactivation can induce cell cycle arrest [44], we measured the expression of EBV genes after treatment with either DMSO or SCD1i+FADS2i. While EBV lytic genes (e.g., BZLF1 and BMRF1) were upregulated after inducible lytic reactivation in P3HR1-ZHT cells, we did not observe any difference in EBV lytic gene expression in LCLs treated with SCD1i+FADS2i (S3 Fig). We observed a decrease in expression of EBNA1 and EBNA3C only after 72 hrs SCD1i+FADS2i treatment, which is likely a byproduct of the growth arrest occurring after 48 hrs desaturase inhibition. Additionally, the expression of EBNA1 is highest in S phase [45], which could explain the reduction in mRNA abundance after SCD1i+FADS2i-mediated arrest at the G1/S transition. Taken together, these data suggest that SCD1 and FADS2 work collectively to provide key desaturase activity throughout EBV-driven B cell outgrowth.

As EBV-mediated B cell transformation and growth mimics stimulatory and costimulatory signals for primary B cells, we next asked whether only EBV-infected B cells required SCD1 and FADS2 activity for proliferation. Treatment with either desaturase inhibitor also decreased the proliferation of B cells stimulated with CpG or CD40L+IL-4 treatment. Simultaneous treatment with both inhibitors further limited proliferation in a manner similar to EBV-infected B cells (Figs 2J–K and S4). Therefore, SCD1 and FADS2 are required for cell division of both uninfected and EBV-infected B cells.

### Genetic knockout of SCD1 and FADS2 suppresses LCL growth

We next sought to complement our pharmacological data implicating SCD1 and FADS2 with a reverse genetic approach. We utilized Cas9-RNP-mediated gene editing to knock out SCD1 and FADS2 in immortalized LCLs (Fig 3A). In addition to transfecting LCLs with an SCD1 or FADS2-targeting guide complex, LCLs were simultaneously transfected with an inert CD46-targeting complex, allowing for precise tracking of successfully transfected LCLs. Successfully gene-edited cells were proxied by CD46-negativity and monitored for ten days post-transfection [13,39]. We also included an MDM2-targeting control, as LCLs require MDM2 to prevent p53-mediated growth arrest and cell death [46]. We found that single SCD1 and FADS2 targeting did not significantly reduce CD46-negative LCL growth over a ten-day culture period (Fig 3B, C). However, dual SCD1 and FADS2 knockout resulted in a decreased percentage of CD46-negative LCLs and decreased proliferation, similar to the MDM2 targeting positive control (Fig 3B–D). The SCD1/FADS2 double knockout population did not survive beyond day ten post-transfection, making it challenging to perform further assays. However, western blotting on day three post-transfection and sequencing of genomic *fads2* after Cas9-RNP transfection confirmed successful targeting of SCD1 and FADS2 in the CD46- LCL population, solidifying the critical role for combined SCD1 and FADS2 activity in EBV-immortalized LCL proliferation (Figs 3E and S5).

### Inhibition of SCD1 and FADS2 sensitizes EBV-immortalized B cells to palmitate-induced lipotoxicity

While palmitate is central to fatty acid metabolism pathways including *de novo* lipid biosynthesis, lipid desaturation, and β-oxidation, excess palmitate can lead to lipotoxicity [47,48]. Canonically, SCD1 converts palmitate and stearate into palmitoleic acid and oleic acid, respectively (Fig 4A). Given the reported ability of FADS2 to metabolize palmitate in

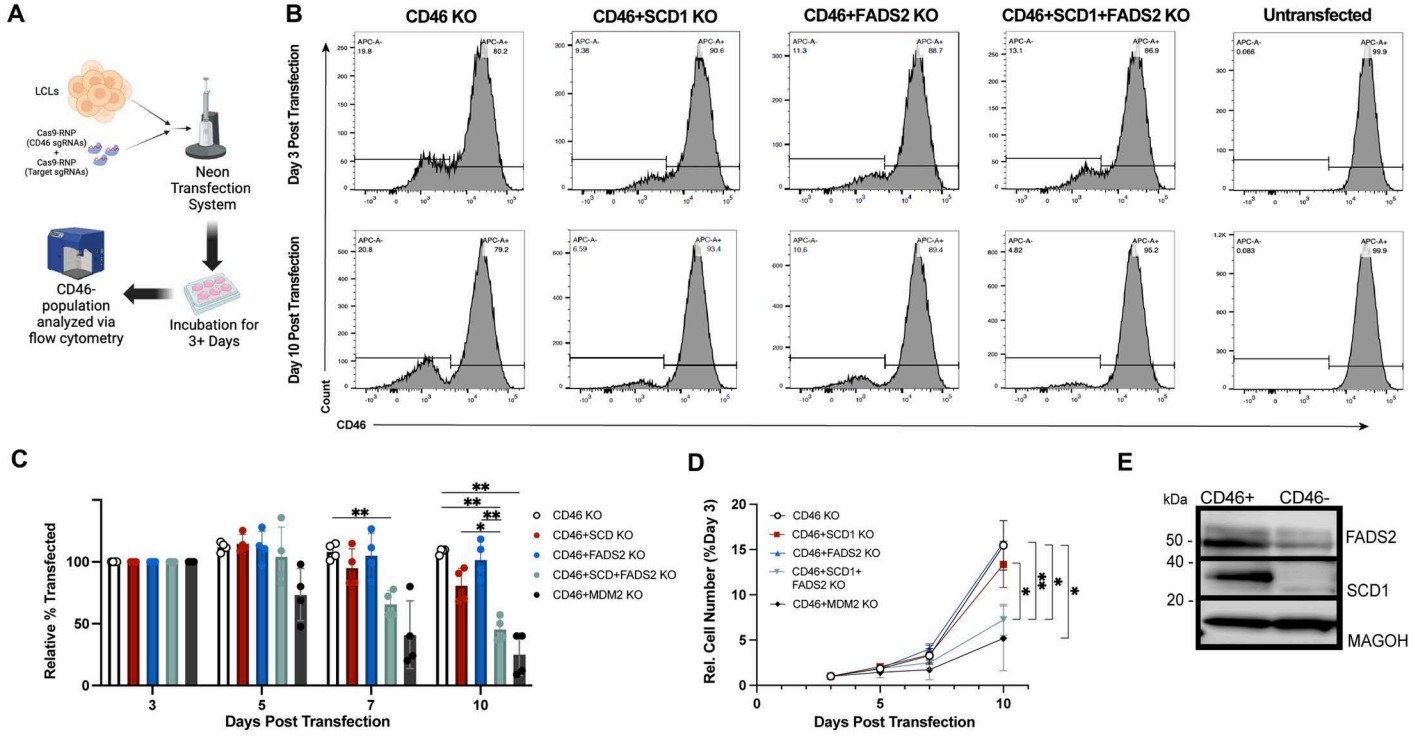

**Fig 3. Genetic knockout of SCD1 and FADS2 impairs LCL proliferation.** (A) Schematic outline for the Cas9-RNP transfection protocol, where CD46 negativity is used as a proxy for transfection. Graphic generated using BioRender. (B) Representative histograms showing % CD46- measured by flow cytometry. (C) DMSO-normalized % transfected (i.e., CD46-) measured by flow cytometry (n = 4, LCLs from 2 donors). All timepoints normalized to day 3 post transfection. (D) Number of CD46- cells over time after targeting with indicated Cas9-RNP complexes. Cell numbers normalized to counting beads and presented relative to starting number at three days post-transfection (n = 4, LCLs from 2 donors). (E) Western blot of transfected LCLs sorted on CD46 expression 3 days post transfection with Cas9-RNP complexes targeting CD46, SCD1 and FADS2. Statistical significance for pairwise comparisons determined using a Tukey's post-hoc test (* p < 0.05, **p < 0.005, ***p < 0.0005, ****p < 0.00005).

SCD1-independent tumors, we sought to test whether the combined SCD1 and FADS2 activity allows LCLs to withstand higher levels of palmitate than the activity of either enzyme alone [31,49].

We first explored the baseline ability of LCLs to tolerate physiologically-relevant concentrations of palmitate [50]. We found that treatment with an increasing dose (from 200-500 μM) of BSA-conjugated palmitate reduced LCL cell number by 96h (Fig 4B). However, only 500 μM palmitate elicited significant cell death (Fig 4C). Rather, the dose-dependent growth reduction induced by palmitate was consistent with a G1/S phase cell cycle arrest (Fig 4D, E).

Given these observations, we hypothesized that SCD1 and FADS2 play a role in minimizing palmitate-induced toxicity in EBV-immortalized LCLs. To test this, we next treated LCLs with increasing concentrations of BSA-palmitate in the presence of either DMSO, SCD1i, FADS2i, or both desaturase inhibitors (SCD1i+FADS2i). All treatments were in the presence of sub-cytotoxic palmitate concentrations (<300 μM). Following 72h of treatment, we observed that single SCD1 or FADS2 inhibition was only cytotoxic in the presence of 200 μM palmitate. In contrast, combined SCD1 and FADS2 inhibition led to increased LCL death at 50 μM palmitate (Fig 4F). Monitoring growth over time, single SCD1 or FADS2 inhibition yielded a complete block in LCL growth in the presence of 100 μM palmitate, which was even more profound following dual desaturase inhibition (Fig 4G). To determine the mechanism of the palmitate-induced cell death, we measured apoptosis by staining for cleaved caspase 3/7 and Annexin V. We found that SCD1 + FADS2 inhibition hypersensitizes LCLs to palmitate-induced apoptosis (Fig 4H). The HDAC inhibitor Panobinostat was used as a positive control for apoptosis because of its potent ability to induce apoptosis in lymphoma cell lines at high concentrations [51]. These data therefore

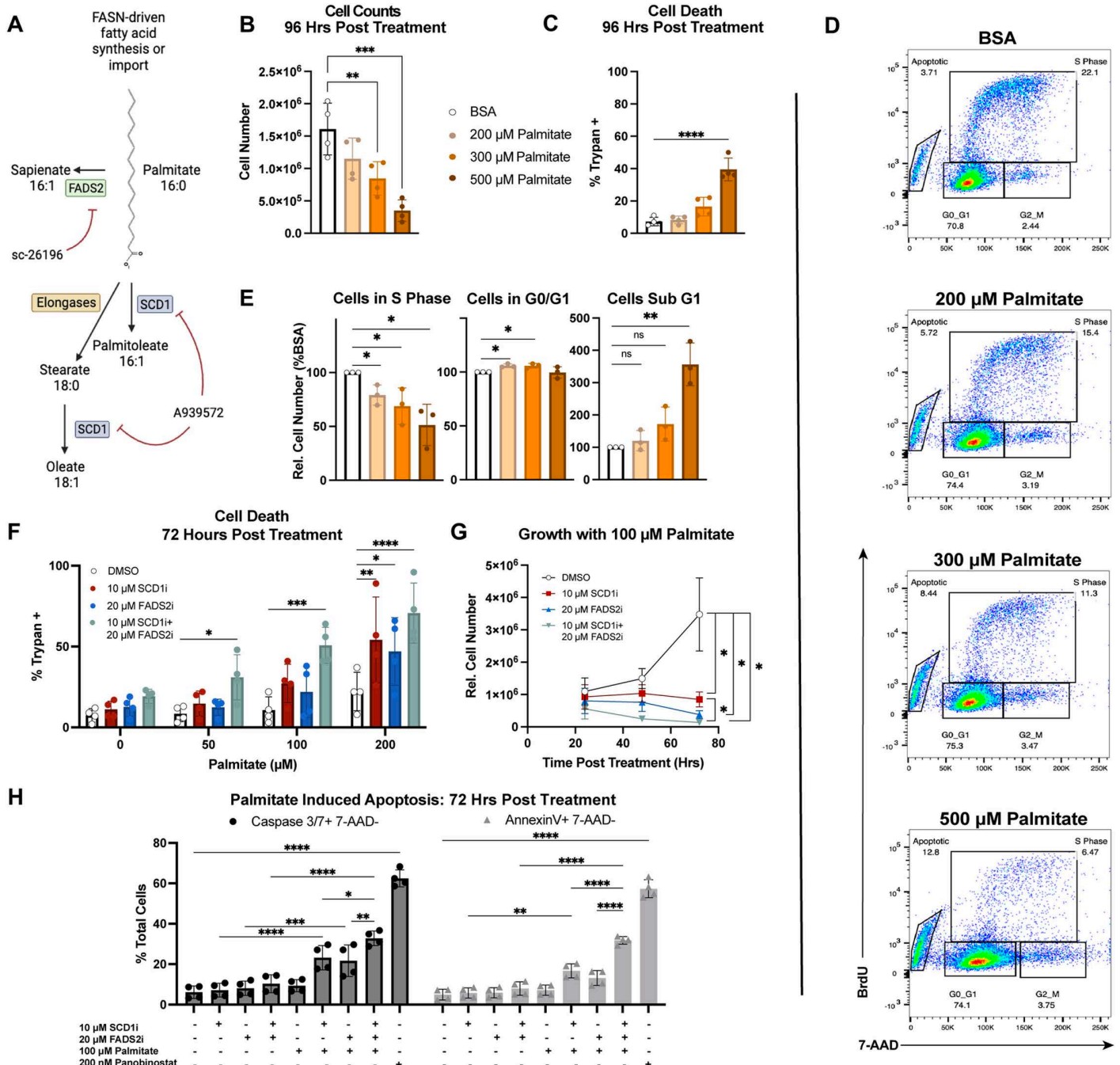

**Fig 4. SCD1 and FADS2 inhibition hypersensitizes LCLs to palmitate-induced lipotoxicity.** (A) Schematic showing how both inhibitors impact the central (i.e., palmitate) fatty acid pathway. Graphic generated using BioRender. (B) Live cell counts after 96h palmitate-BSA treatment using Trypan dye (n = 4, LCLs from 2 donors). (C) Percentages of dead cells after treatment with palmitate-BSA for 96h, determined using Trypan dye (n = 4, LCLs from 2 donors). (D) Representative dot plots showing BrdU incorporation (FITC) against total DNA (7-AAD) after 48h palmitate-BSA treatment. (E) Relative percentage of cells in each cell cycle phase after palmitate-BSA treatment for 48h normalized to BSA control (n = 3, LCLs from 3 donors). (F) Percentages of dead cells after treatment with DMSO, 10 μM A939572 (SCD1i), 20 μM SC-26196 (FADS2i), or both inhibitors and indicated concentration of palmitate-BSA for 72h (n = 4, LCLs from 2 donors). Cell death measured as in C. (G) Growth curve showing how desaturase inhibitors affect LCL growth in the presence of palmitate-BSA (n = 4, LCLs from 2 donors). Growth was measured using Cell Titer Glo. (H) Percentages of early apoptotic cells (7-AAD-, and either cleaved caspase 3/7+ or Annexin V+) after 72h treatment (n = 4, LCLs from 2 donors). 200 nM Panobinostat used as a positive control for apoptosis. Statistical significance for pairwise comparisons determined using a Tukey's post-hoc test (* p < 0.05, **p < 0.005, ***p < 0.0005, ****p < 0.00005).

reveal a metabolic vulnerability of EBV-infected LCLs, where simultaneous SCD1 and FADS2 inhibition hypersensitizes LCLs to apoptosis from physiologically-relevant concentrations of palmitate [50].

### SCD1 and FADS2 are key regulators of growth-promoting metabolism in LCLs

Palmitate induces lipotoxicity in cells by altering the saturated/unsaturated fatty acid (SFA/UFA) ratio. In many cancers such as breast, colorectal, pancreatic, and liver cancer, the saturated/monounsaturated fatty acid (SFA/MUFA) ratio is commonly dysregulated [52], suggesting that increased SCD1 activity might be a biochemical hallmark of tumorigenesis that prevent SFA accumulation and promote growth. Given these considerations, we wondered whether tilting the SFA/MUFA balance back in favor of MUFAs might rescue the cytotoxicity phenotype induced by the SFA palmitate. To do this, we added exogenous BSA-conjugated oleate, an SCD1-derived MUFA, alongside palmitate in the presence of the SCD1 and FADS2 inhibitors. BSA-conjugated oleate fully rescued the loss of viable LCLs driven by palmitate in the presence of SCD1 and/or FADS2 inhibitors, albeit not to levels observed in the absence of the inhibitors (Fig 5A). Oleate rescue was due to suppression of palmitate-induced apoptosis, but not growth arrest associated with SCD1 + FADS2 inhibition (Fig 5A–C). We reasoned that this may be due to the growth arrest phenotype not being the result of palmitate accumulation.

To test this hypothesis, we measured free fatty acid levels using fatty acid methyl esterification (FAME) and mass spectrometry-based targeted free fatty acid profiling that could measure palmitate levels after desaturase inhibition. We did not observe a significant increase in saturated fatty acid (SFA) levels after desaturase inhibition, except for behenate (docosanoate). Instead, we observed a more discriminatory trend towards increased free unsaturated fatty acid levels after 72h of combined SCD1 + FADS2 inhibition, relative to SFA levels. Precisely, we observed significant accumulation of linoleate (LA), linolelaidate, cis-10-pentadecenoate, and docosahexaenoic acid (DHA). (Figs 5D, E and S6). Thus, these data suggest that simultaneous SCD1 + FADS2 inhibition induces a broad shift in cellular fatty acid metabolism, likely involving alternate synthesis pathways, in order to maintain intracellular pools of PUFAs.

To gain a more mechanistic understanding of the SCD1/FADS2-inhibitor associated growth arrest, we measured oxidative phosphorylation and glycolysis via oxygen consumption rates (OCR) and extracellular acidification rates (ECAR), respectively. We observed that SCD1 + FADS2 inhibition reduced basal oxidative phosphorylation, basal glycolysis, the ability for LCLs to resort to glycolysis after electron transport chain inhibition, and the proportion of ATP arising from oxidative phosphorylation (Fig 5F–I). Together, these data suggest that SCD1 and FADS2 are key regulators of SFA/UFA ratios and, likely by extension, growth-promoting metabolism in LCLs.

### Inhibition of SCD1 and FADS2 suppresses growth of EBV+ lymphoma cells

Finally, we sought to investigate the role of SCD1 and FADS2 in EBV+ tumor cells using the AIDS-associated immunoblastic lymphoma cell line, IBL-1 [53]. Pharmacological inhibition of SCD1 and FADS2 led to a significant reduction of IBL-1 growth, similar to what we observed in LCLs (Fig 6A). Likewise, knocking out either SCD1 or FADS2 individually led to a significant reduction of CD46-negative proxy IBL-1 cells at 7 and 14 days post-transfection, with SCD1-loss being less well tolerated than FADS2 loss at 14 days post-transfection (Fig 6B). Combined knockout of SCD1 and FADS2 in IBL-1 cells was not well tolerated, further establishing a cooperative role for SCD1 and FADS2 in promoting EBV-infected B cell growth not only in LCLs but in viral lymphomas (Fig 6B). Finally, we explored whether the presence of SCD1 and FADS2 protected IBL-1 cells from palmitate-driven lipotoxicity. Similar to what was observed in LCLs, we found that dual SCD1/FADS2 inhibition rendered IBL-1 cells hyper-sensitive to palmitate (Fig 6C). These data, therefore, highlight a proof-of-concept for pursuing the SFA:MUFA ratio as a novel therapeutic target in EBV+ lymphomas.

## Discussion

In order for EBV to transform resting B cells into rapidly-dividing, tumorigenic LCLs, EBV must drive shifts in metabolism to promote cell division. As fatty acids are required for the synthesis of new cell membranes, fatty acid biosynthesis is

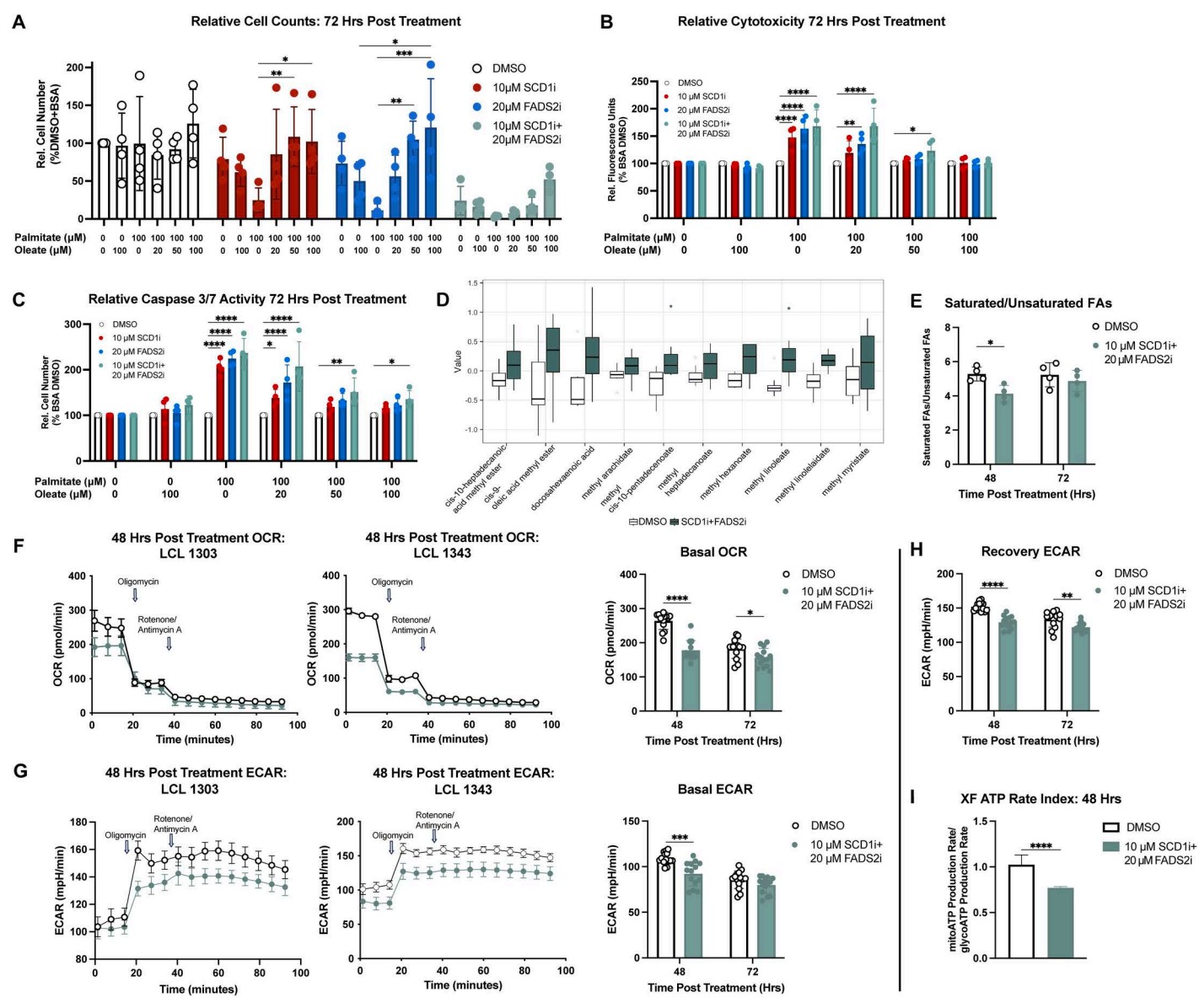

**Fig 5. Growth arrest driven by desaturase inhibition is associated with broad metabolic shifts.** (A) Relative LCL numbers after treatment with DMSO, 10 μM A939572 (SCD1i), 20 μM SC-26196 (FADS2i), or both inhibitors and indicated concentrations of palmitate-BSA and/or oleate-BSA for 72h (n = 4, LCLs from 2 donors). Growth was measured using CellTiter Glo. (B) Cytotoxicity (normalized to DMSO+BSA control) after 72h treatment with desaturase inhibitors, palmitate-BSA, and/or oleate-BSA (n = 4, LCLs from 2 donors). Cytotoxicity was measured using CellTox Green DNA dye. (C) Caspase 3/7 activity (normalized to DMSO+BSA control) after 72h treatment with desaturase inhibitors, palmitate-BSA, and/or oleate-BSA (n = 4, LCLs from 2 donors). Caspase 3/7 activity was measured using Caspase 3/7 Glo. (D) Boxplots comparing normalized abundances between DMSO-treated and SCD1i+FADS2i-treated LCLs (n = 4, LCLs from 4 donors). Only the top ten most differentially abundance fatty acid species are shown.(E) Ratios of saturated: unsaturated free fatty acid concentrations after indicated treatments (n = 4, LCLs from 4 donors). (F) Oxygen consumption rates (OCR) over time in LCLs from two donors, along with basal OCR (n = 14, LCLs from 2 donors). (G) Extracellular acidification rates (ECAR) over time in LCLs from two donors, along with basal ECAR (n = 14, LCLs from 2 donors). (H) ECAR measurements between oligomycin and rotenone/antimycin A injections, showing extent to which metabolism shifts towards glycolysis after electron transport chain inhibition (n = 14, LCLs from 2 donors). (I) Ratio of ATP produced from oxidative phosphorylation to ATP produced from glycolysis after 48h of indicated treatments (n = 14, LCLs from 2 donors). Statistical significance for pairwise comparisons was determined using either a Student's unpaired T test or Tukey's post-hoc test (* $p < 0.05$, ** $p < 0.005$, *** $p < 0.0005$, **** $p < 0.00005$).

PLOS Pathogens

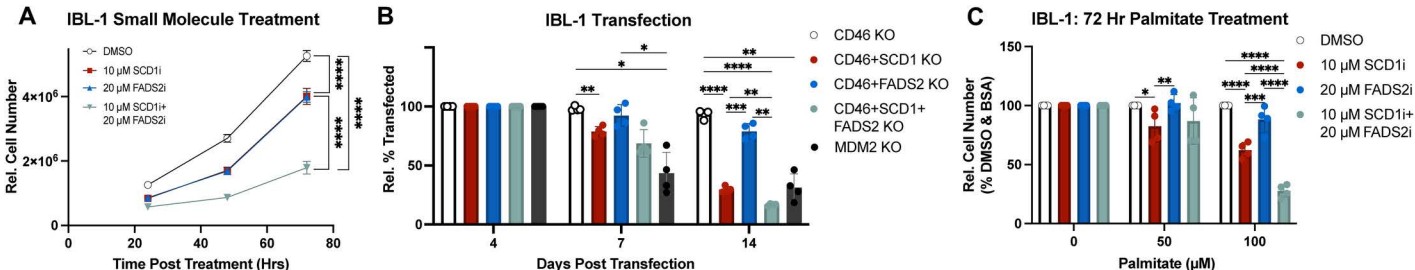

**Fig 6. IBL-1 cells are sensitive to growth arrest after SCD1 and FADS2 inhibition.** (A) Growth curve of IBL-1 cells treated with desaturase inhibitors (n = 4). Growth was measured using Cell Titer Glo. (B) DMSO-normalized % transfected IBL-1 cells (i.e., CD46-) measured by flow cytometry (n = 4). (C) Growth curve of IBL-1 cells treated with desaturase inhibitors (n = 4). Growth was measured as in A. Statistical significance for pairwise comparisons determined using a Tukey's post-hoc test (* p < 0.05, **p < 0.005, ***p < 0.0005, ****p < 0.00005).

crucial for most cancers. However, the precise role of fatty acids can be fine-tuned through elongation and desaturation pathways, generating diverse fatty acid species [14]. Since fatty acid desaturation has been implicated in other cancer types, we sought to determine whether fatty acid desaturation is an important metabolic pathway for EBV-mediated B cell transformation. Our findings reveal that fatty acid desaturation is indeed crucial for EBV-mediated B cell division, as either the SCD1 or the FADS2-driven fatty acid desaturation pathway is required for LCL growth.

We began by characterizing the role of fatty acid desaturases in the proliferation of early EBV-infected B cells. Given their expression during the early stages of infection, we identified that SCD1 and FADS2 were induced by the EBV master transcriptional regulator EBNA2, which is equally responsible for upregulating other metabolic pathways crucial for EBV-driven B cell outgrowth [11,12]. EBNA2 regulation of SREBP2 supports mevalonate metabolism, supporting an EBNA2-driven, SREBP2-mediated mechanism of SCD1 induction post-EBV infection [11]. FADS2 expression is additionally regulated transcriptionally by peroxisome proliferator-activated receptor-alpha (PPAR-α) [21]. However, gene expression changes in EBV-infected B cells did not show marked changes in PPAR-α expression following EBV infection, suggesting PPAR-α-mediated FADS2 regulation is unlikely in EBV-infected B cells [54]. We focused our study on SCD1 and FADS2 due to their co-implication in cancer etiology [55].

Next, we used pharmacological inhibitors to determine the role of SCD1 and FADS2 in EBV-driven B cell growth and survival, considering their increased expression following EBV infection and through B cell outgrowth. We found that dual inhibition of SCD1 and FADS2 led to G1/S-phase cell cycle arrest, confirming their importance in maintaining proliferation both early after infection and in immortalized LCLs. Inhibition of SCD1 and FADS2 also reduced the proliferation of stimulated uninfected B cells, which is consistent with previous work demonstrating that stimulated B cells require SCD1-generated MUFAs for growth [56]. These data point towards a more ubiquitous nature in promoting B cell proliferation. However, the significantly higher expression of both SCD1 and FADS2 in EBV-immortalized LCLs still positions these desaturases as viable therapeutic targets against EBV-driven lymphomas. Future studies could further explore other metabolic vulnerabilities induced by desaturase inhibition to minimize off-target effects and maximize therapeutic benefit.

Genetic knockout of both SCD1 and FADS2 using gene-targeting Cas9-RNP complexes displayed similar effects on cell growth. Considering the genetic experiments demonstrate significantly less cell growth with simultaneous knockout compared to individual desaturase knockout, and our pharmacological data demonstrate that the SCD1 and FADS2 inhibitors work synergistically, the two desaturases may compensate for each other. A compensatory role for FADS2 in the context of SCD1 inhibition has been established in previous studies, precisely through the production of the monounsaturated fatty acid sapienate from palmitate instead of palmitoleate [31,32]. While we did not measure sapienate in our study, it is possible other desaturases like FADS1 might be compensating for SCD1 and FADS2 loss through a similar

alternate pathway to prevent cell death. Future studies could identify such metabolic pathways to target alongside SCD1 and FADS2 in EBV-associated lymphomas.

While dual SCD1 and FADS2 pharmacological inhibition was not lethal over the course of treatment times studied, we observed that these cells were hypersensitive to palmitate compared to either single SCD1/FADS2-treated or untreated cells. Our observations of palmitate-induced lipotoxicity at ~200μM match what has been found in a previous study [57]. Early after EBV infection, EBV induces fatty acid synthase (FASN) expression through EBNA2, and later during LCL outgrowth through Latent Membrane Protein 1 (LMP1) [58,59]. FASN primarily drives *de novo* synthesis of palmitate. Consistent with the EBV-induced upregulation of FASN during infection, low enough concentrations of palmitate (25μM) promoted LCL growth [59]. This observed expression of FASN throughout EBV-driven B cell immortalization, combined with the observed cytotoxicity of palmitate upon dual SCD1 and FADS2 loss, led us to hypothesize that SCD1 and FADS2 prevent palmitate-induced lipotoxicity throughout infection. However, we did not observe an increase in palmitate levels within the free fatty acid pool after SCD1 + FADS2 inhibition, suggesting that the cell cycle arrest and reduction in growth-promoting metabolism associated with SCD1 + FADS2 inhibition are not directly due to palmitate accumulation.

Instead, we hypothesize that LCLs utilize other pathways to increase the pool of free unsaturated fatty acids after the inhibition of two key desaturases, including shunting saturated fatty acids to other desaturases or lipid synthesis enzymes (e.g., FADS1 or DGAT1, respectively), lipolysis of complex lipids, or intake of more unsaturated fatty acids from the media. Many differentially abundant fatty acid species are unable to be synthesized by human cells (e.g., cis-10-heptadecenoate and linoleate) [60]. Therefore, the increase in abundance of those species after desaturase inhibition must stem from either more fatty acid import or lipolysis. In particular, lipase activity has been linked to regulating the pool of non-esterified fatty acids after desaturase inhibition. A recent study revealed that SCD1 inhibition in CD4 + T cells results in an increase in free PUFA abundance (e.g., DHA) due to lipolysis driven by increased adipose triglyceride lipase (ATGL) activity [61]. As we also observed an increase in non-esterified DHA abundance after SCD1/FADS2 inhibition, it is possible that desaturase inhibition in LCLs induces similar lipase activity. Additionally, since the excessive lipolysis and release of PUFAs from lipid droplets can cause redox imbalance and mitochondrial dysfunction, it is also possible that the increase in abundance of non-esterified linoleate or DHA after desaturase inhibition drives the cell cycle arrest [62]. As PUFAs esterified as phosphatidylcholines or phosphatidylethanolamines are involved in ferroptosis (a type of cell death mediated by the reaction of ROS with PUFAs) [63], any lipolysis of membrane lipids may help prevent ferroptosis, especially considering the observed mitochondrial dysfunction after desaturase inhibition. Therefore, the ability for LCLs to regulate the free fatty acid pool after SCD1/FADS2 inhibition may be a survival strategy. Treating LCLs with desaturase inhibitors and extra palmitate may overwhelm the machinery used to adjust the saturated/unsaturated fatty acid ratios, resulting in apoptosis. The growth arrest phenotype may be due to metabolites being redistributed in such a way that allows LCLs to avoid cell death while blocking further cell division.

While high doses of palmitate are implicated in lipotoxicity and apoptosis in a variety of cell types, the exact mechanism for palmitate-induced cell cycle arrest and apoptosis remains largely undefined [48,52]. Studies suggest that palmitate accumulation results in endoplasmic reticulum (ER) stress, with an upregulation of CHOP and spliced XBP1, and a halting of protein translation that can be rescued by oleate cotreatment [64–66]. However, EBV latency alone inhibits the production of spliced XBP1 by the ribonuclease IRE1, as the spliced form can promote lytic reactivation by activating the Rta and Zta promoters [54,67]. Due to this altered cellular context, it is likely that other mechanisms of palmitate-induced lipotoxicity also contribute to the observed phenotype, especially at the intermediate palmitate doses. As oleate is implicated in specifically rescuing palmitate-induced ER stress, the involvement of other pathways would be consistent with our observation that oleate was unable to rescue growth arrest after desaturase inhibition. Palmitate treatment is also known to induce mitochondrial fragmentation, resulting in reduced ATP production via oxidative phosphorylation and, consequentially, increased glycolysis [68,69]. Given previous studies' findings that aerobic glycolysis is important for EBV-mediated B cell proliferation, it is possible that palmitate-induced changes in mitochondrial performance (after incorporation into

membrane phospholipids) may drive the growth arrest seen after treatment with intermediate palmitate doses [9,13]. Additional mechanisms of palmitate-induced lipotoxicity observed in some cell types include redox imbalance, increased lysophosphatidylcholine synthesis, and lysosomal $Ca^{2+}$ leakage [69–72]. Therefore, it is likely that the sensitivity of EBV-infected B cells to palmitate is due to a combination of many factors.

Finally, our study assessed whether the effects of SCD1 and FADS2 perturbation observed *in vitro* in EBV-immortalized LCLs held true in tumor-derived models of EBV+ lymphomas, thereby pointing to potential therapeutic applications. We found that similar to LCLs, combined SCD1 and FADS2 inhibition arrested growth and hypersensitized EBV+ AIDS immunoblastic lymphoma IBL-1 cells to increased palmitate, pointing towards a unique targetable mechanism in EBV-associated lymphomas. Future studies would need to establish preclinical models of EBV-associated lymphomas to further validate this approach and demonstrate that the *in vitro* effects of dual SCD1 and FADS2 inhibition on cell growth translate to *in vivo* models. These studies would also need to adequately account for the fact that SCD1 is ubiquitously expressed and could, thus, lead to undesirable side effects. Complementary studies that explore the flow of palmitate, such as isotope-labeled palmitate tracing, could further shed light on the involvement of SCD1 and FADS2 in palmitate metabolism throughout infection and in EBV-transformed tumors, thereby informing future therapeutic approaches that minimize adverse side effects.

## Supporting information

**S1 Fig. Replicate Western blots showing EBV-driven upregulation of SCD1 and FADS2.**
(TIF)

**S2 Fig. Dose response data and isobolograms for SCD1i+FADS2i.** (A) Dose-response data for SCD1i. Relative cell number measured using CellTiter Glo and normalized to DMSO. (B) Dose-response data for FADS2i. Relative cell number measured using CellTiter Glo and normalized to DMSO. (C-D) Isobologram at 48 hrs and 72 hrs, respectively, using IC50 for SCD1i on y axis and sub-IC50 (maximum soluble concentration) for FADS2i on x axis. Plotted points show IC50 values for SCD1i in the presence of that concentration of FADS2i. (E) Drug combination indices (CI) at indicated dose of FADS2i. A CI value less than 1 indicates synergism. Statistical significance for pairwise comparisons determined using an unpaired Student's two-tailed T test (* $p < 0.05$, ** $p < 0.005$, *** $p < 0.0005$, **** $p < 0.00005$).
(TIF)

**S3 Fig. Desaturase-dependent growth arrest does not involve changes in EBV gene expression.** (A-E) qPCR data showing relative expression of latency-associated EBV genes, relative to DMSO. *Setdb1* used as a housekeeping gene for normalization. (F-I) qPCR data showing relative expression of lytic-associated EBV genes, relative to DMSO. *Setdb1* used as a housekeeping gene for normalization. The ZHT cell line treated with 4-hydroxytamoxifen (4HT) to induce lytic reactivation was included as a positive control. Statistical significance for pairwise comparisons determined using an unpaired Student's two-tailed T test (* $p < 0.05$, ** $p < 0.005$, *** $p < 0.0005$, **** $p < 0.00005$).
(TIF)

**S4 Fig. Primary B cells are sensitive to SCD1+FADS2 inhibition.** (A) Representative flow cytometry histograms five days post stimulation, showing CellTrace Violet dilution as cells divide. Counts are normalized to mode. (B) Graphs showing the number of B cells that proliferated (CD19+/CTV$^{lo}$) three days after treatment and stimulation with CpG. Cell numbers normalized to counting beads and presented as a percentage of DMSO-treated controls (C) Graphs showing the number of B cells that proliferated (CD19+/CTV$^{lo}$) three days after treatment and stimulation with recombinant CD40L and IL-4. Cell numbers normalized to counting beads and presented as a percentage of DMSO-treated controls. Statistical significance for pairwise comparisons determined using a Tukey's post-hoc test (* $p < 0.05$, ** $p < 0.005$, *** $p < 0.0005$, **** $p < 0.00005$).
(TIF)

**S5 Fig. Targeting FADS2 with Cas9-RNP complexes does not fully deplete FADS2 levels, but sgRNAs are on-target.** (A) Quantification of Western blot in Fig 3E, normalized to MAGOH and WT. (B) Discordance of FADS2 genomic sequences between WT and CD46＋FADS2 KO LCLs (left), and estimated percentage of each indel in mixture (right). Graphs generated using Synthego's Inference of CRISPR Edits tool. (C) Genomic FADS2 sequences in WT and FADS2-KO LCLs. Black lines indicate binding sites of FADS2 sgRNAs. Figure generated using Synthego's Inference of CRISPR Edits tool.
(TIF)

**S6 Fig. Analysis of fatty acid methyl esterification (FAME) free fatty acid profiling data.** (A) Heatmap showing Pearson's correlation coefficient values between pairs of fatty acid species. (B) Plot showing free fatty acid species with significant levels of enrichment in samples treated with 10 µM SCD1i+20 µM FADS2i. P values were determined using limma analysis of normalized FAME free fatty acid profiling data. A negative log2 fold change indicates enrichment in the SCD1i+FADS2i-treated samples, while a positive log2 fold change indicates enrichment in the DMSO-treated samples.
(TIF)

**S1 File. Zipped data corresponding to Fig 1** . (A) Raw data corresponding to Fig 1C. (B) Uncropped Western blots corresponding to Fig 1E. (C) Raw data corresponding to Fig 1F.
(ZIP)

**S2 File. Zipped data corresponding to Figs 2 and S4.** (A) Raw data corresponding to Fig 2A, B. (B) Raw data corresponding to Fig 2C–D. (C) Raw data corresponding to Fig 2E. (D) Raw data corresponding to Fig 2F–G. (E) Raw data corresponding to Fig 2H. (F) Raw data corresponding to Fig 2I. (G) Raw data corresponding to Figs 2J, K and S4.
(ZIP)

**S3 File. Zipped data corresponding to Fig 3.** (A) Gating strategy corresponding to flow cytometry data displayed in Fig 3B, C. (B) Complete flow cytometry dot plots at 3 days post transfection, corresponding to Fig 3B, C. (C) Complete flow cytometry dot plots at 10 days post transfection, corresponding to Fig 3B, C. (D) Raw data corresponding to Fig 3D. (E) Uncropped Western blot corresponding to Fig 3E.
(ZIP)

**S4 File. Zipped data corresponding to Fig 4.** (A) Raw data corresponding to Fig 4B, C. (B) Gating strategy corresponding to flow cytometry data in Fig 4D, E. (C) Complete flow cytometry dot plots corresponding to Fig 4D, E. (D) Raw data corresponding to Fig 4D, E. (E) Raw data corresponding to Fig 4F. (F) Raw data corresponding to Fig 4G. (G) Gating strategy for Annexin-V measurement, corresponding to Fig 4H. (H) Gating strategy for cleaved caspase 3/7 measurement, corresponding to Fig 4H. (I) Raw data corresponding to Fig 4H.
(ZIP)

**S5 File. Zipped data corresponding to Figs 5 and S6.** (A) Raw data corresponding to Fig 5A–C. (B) Raw data corresponding to Figs 5D, E and S6. (C) Raw data corresponding to Fig 5F. (D) Raw data corresponding to Fig 5G, H. (E) Raw data corresponding to Fig 5I.
(ZIP)

**S6 File. Zipped data corresponding to Fig 6.** (A) Raw data corresponding to Fig 6A and 6C. (B) Raw data corresponding to Fig 6B.
(ZIP)

**S7 File. Zipped data corresponding to supplementary figures.** (A) Uncropped Western blot corresponding to S1 Fig. (B) Raw data corresponding to S2 Fig. (C) Raw data corresponding to S3 Fig.
(ZIP)

## Acknowledgments

We would like to acknowledge the members of the Luftig and Hirschey labs for their help and support, specifically Nico Reinoso for providing experimental support in response to reviewer critiques. We would also like to acknowledge the University of North Carolina Chapel Hill lipidomics core – especially Brandie Ehrmann – for conducting the FAME free fatty acid profiling analysis.

## Author contributions

**Conceptualization:** Emmanuela N Bonglack, Kaeden K Hill, Micah A Luftig.

**Data curation:** Emmanuela N Bonglack, Kaeden K Hill, Ashley P Barry, Alexandria Bartlett.

**Formal analysis:** Emmanuela N Bonglack, Kaeden K Hill, Alexandria Bartlett, Pol Castellano-Escuder, Matthew D Hirschey.

**Funding acquisition:** Emmanuela N Bonglack, Micah A Luftig.

**Investigation:** Emmanuela N Bonglack, Kaeden K Hill, Ashley P Barry, Alexandria Bartlett.

**Methodology:** Pol Castellano-Escuder.

**Project administration:** Micah A Luftig.

**Resources:** Micah A Luftig.

**Supervision:** Emmanuela N Bonglack, Micah A Luftig.

**Validation:** Emmanuela N Bonglack, Kaeden K Hill, Alexandria Bartlett, Matthew D Hirschey.

**Visualization:** Emmanuela N Bonglack, Kaeden K Hill, Matthew D Hirschey.

**Writing – original draft:** Emmanuela N Bonglack, Kaeden K Hill.

**Writing – review & editing:** Emmanuela N Bonglack, Kaeden K Hill, Ashley P Barry, Alexandria Bartlett, Matthew D Hirschey, Micah A Luftig.

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
