## [Decision Letter · Decision Letter 0]

16 Dec 2024

Fatty acid desaturases link cell metabolism pathways to promote proliferation of Epstein-Barr virus-infected B cells

PLOS Pathogens

Dear Dr. Luftig,

Thank you for submitting your manuscript to PLOS Pathogens. After careful consideration, we feel that it has merit but does not fully meet PLOS Pathogens's publication criteria as it currently stands. Therefore, we invite you to submit a revised version of the manuscript that addresses the points raised during the review process.

Please submit your revised manuscript within 60 days Feb 14 2025 11:59PM. If you will need more time than this to complete your revisions, please reply to this message or contact the journal office at plospathogens@plos.org. Please include the following items when submitting your revised manuscript:

We look forward to receiving your revised manuscript.

Kind regards,

Benjamin E Gewurz, M.D., Ph.D.

Academic Editor

PLOS Pathogens

Alison McBride

Section Editor

PLOS Pathogens

Editor-in-Chief

PLOS Pathogens

orcid.org/0000-0003-2946-9497

Michael Malim

Editor-in-Chief

PLOS Pathogens

orcid.org/0000-0002-7699-2064

**Additional Editor Comments :**

Thank you for your patience while we acquired reviews from 3 experts in the field. As you can see below, each appreciated merits of the work, but have a number of specific comments to address for a revision, in particular the point raised by reviewer #1 and #2's about the specificity of the observation to EBV (vs B-cells more generally) and the request to test phenotypes in LCLs from multiple donors. Addressing a number of other points raised should be relatively straightforward and would be helpful.

**Journal Requirements:**

1) Please provide an Author Summary. This should appear in your manuscript between the Abstract (if applicable) and the Introduction, and should be 150-200 words long. The aim should be to make your findings accessible to a wide audience that includes both scientists and non-scientists. Sample summaries can be found on our website under Submission Guidelines:

https://journals.plos.org/plospathogens/s/submission-guidelines#loc-parts-of-a-submission

- TM on page: 9.

4) We notice that your supplementary Figure is included in the manuscript file. Please remove it and upload it with the file type 'Supporting Information'. Please ensure that each Supporting Information file has a legend listed in the manuscript after the references list.

5) We note that your Data Availability Statement is currently as follows: "All relevant data are within the manuscript and its Supporting Information files."  Please confirm at this time whether or not your submission contains all raw data required to replicate the results of your study. Authors must share the “minimal data set” for their submission. PLOS defines the minimal data set to consist of the data required to replicate all study findings reported in the article, as well as related metadata and methods (https://journals.plos.org/plosone/s/data-availability#loc-minimal-data-set-definition).

**Reviewers' Comments:**

Reviewer's Responses to Questions

**Part I - Summary**

Reviewer #1: The manuscript focuses on the role of fatty acid desaturases in B cells transformed by EBV infection in vitro. While the role of SCD1 and FADS2 have not been directly investigated in the context of EBV LCLs before, the observations made in the presented manuscript are largely expected based on published studies that use other transformed/cancer cell lines. It is also not clear whether the observed effects are also extendable to primary B cells, given that most primary B cell subsets express significantly higher SCD1 levels as compared to other immune cell types, with some expression of FADS2 also observed.

Reviewer #2: In the manuscript by Bonglack et. al. the authors use previous RNAseq and proteomic data from EBV infected B-cell outgrowth experiment to show that two fatty acid desaturases are strongly increased following B-cell infection and outgrowth, SCD1 and FADS2. Drug inhibition of these desaturases together lead to inhibition of infected B-cell proliferation as demonstrated by a decrease of cells in S phase. They use a CRISPR/Cas9 co-knockout of SCD1 and FADS2 to show similar decreases in proliferation which is the only evidence that the inhibitor findings are not off-target effects from the inhibitors used. The fatty acid desaturase inhibitors also sensitize the cells to exogenously added palmitate toxicity as indicated by increased apoptosis. Added unsaturated fatty acids relieve the palmitate toxicity showing that the increased apoptosis is likely due to a saturated/unsaturated imbalance. As the mode of inhibition is different for the palmitate toxicity, it is not clear this is relevant to the decreased LCL outgrowth by the inhibitors, which is not clearly described in the discussion. They perform lipid analysis of the cells with and without the inhibitors and oddly see a decrease in the saturated to unsaturated fatty acid levels which is not expected, does not make sense given the conclusions of the manuscript and is poorly described in the discussion. They find that inhibition of the desaturases leads to decreased oxygen consumption rate and decreased basal and recovery extracellular acidification rates leading to the conclusion that the imbalance of unsaturated to saturated fatty acids leads to inhibition of anabolic metabolism. This result is confounded by the actual increase in unsaturated fatty acids in the cells with the inhibitors. Finally, they find similar results in a clinically relevant lymphoma line. Overall, the findings are highly interesting and the experiments are relatively convincing. However, they poorly describe the relationship between different experiments that are at odds with some of their conclusions as detailed below.

Reviewer #3: In this manuscript, the authors explore the role of fatty acid desaturases in the Epstein-Barr virus mediated outgrowth of infected B cells. This study systematically evaluates how stearoyl Co-A desaturase (SCD1) and fatty acid desaturase 2 (FADS2) levels and activity are increased by EBV infection and subsequently impacts proliferation and cell cycle arrest. This study demonstrates that dual inhibition or dual depletion of SCD1 and FADS2 reduces B cell outgrowth. Their results suggest that SCD1 and FADS2 provide redundant functions in support of EBV-supported proliferation of B cells, including EBV+ IBL-1 cells. This is an intriguing observation that may have broader implications. Further, the authors performed some elegant studies feeding saturated palmitate (the substrate for SCD1/FADS2) and one of their unsaturated products (oleate) to demonstrate that the balance of saturated/unsaturated fatty acid content of the cell is necessary for EBV-mediated cell growth. The manuscript is clearly written. The experiments were conducted systematically with appropriate controls. The conclusions have been appropriately drawn. The findings will be of significant interest to the EBV field, and likely the broader virology field as other viruses may also influence the balancing saturated and unsaturated fatty acid content of cells.

**Part II – Major Issues: Key Experiments Required for Acceptance**

Reviewer #1: The key observations in the presented study should be tested in the context of primary uninfected B cells stimulated to proliferate physiologically (BCR crosslinking, CD40 or TLR ligation, etc) to determine the extent to which the observed phenotypes are modified by EBV.

As an extension of the comment above, observations presented in Fig. 4 are consistent with the fundamental biochemistry of SCD1 and FADS2 and are expected to be observed in most transformed cells and, perhaps, even the primary B cells stimulated to proliferate via physiological stimuli.

It is not clear whether the key phenotypes were tested in LCLs derived from more than one independent donors. The inter-donor variability should be shown.

It is not clear whether the experimental manipulation of SCD1 and FADS2 activity and expression has an effect on EBV gene expression and latent/lytic cycle

Reviewer #2: 1. The inhibitors have strong effects on the proliferation of the cells as do the knockouts. However, is this actually an effect specific to EBV infected B-cells. There are no controls for the general effect of these inhibitors on the proliferation of B-cells. They state that the desaturases are relatively ubiquitous in human cells. While they are increased in EBV infected cells, the effect of these inhibitors should be shown in uninfected cells.

2. They state in the discussion line 550, “the observed cytotoxicity of palmitate upon dual SCD1 and FADS2 loss, suggests SCD1 and FADS2 prevent palmitate-induced lipotoxicity throughout infection” This actually is not borne out by the data. While the desaturases can prevent palmitate lipotoxicity, the data indicates that this does not occur in the cells they test as the inhibition of SCD1 and FADS2 leads to inhibition of proliferation not cell death. If they were protecting form lipotoxicity there would also be increased cell death. This is actually discussed later in the discussion when they point out that palmitate is not increased when they examine lipid species, further indicating that at least in the conditions tested in the manuscript, protecting from palmitate toxicity is not a critical mechanism for the desaturases. This should be properly discussed.

3. The authors need to provide a stronger explanation for why there is a decreases saturated to unsaturated fatty acid profile upon inhibition of the desaturases as this is directly counter to what would be expected.

Reviewer #3: None.

**Part III – Minor Issues: Editorial and Data Presentation Modifications**

Reviewer #1: SCD1 and FADS2 regulation of fatty acid saturation has been linked to ferroptosis, ER stress response, and eicosanoid signaling, in addition to the expected link to metabolism. While ER stress is touched upon in the discussion, the discussion needs to be fleshed out to include additional potential mechanisms and their fit with the observed phenotypes.

Apoptosis assays in Fig. 2H do not have a positive control

Fig. 3 traces proportion of targeted LCLs in a mixed cell culture over 10 days. The conclusion that is made refers to decreased cell proliferation in targeted cells, but there are no data to support this claim, as multiple factors can affect the change in proportion of targeted vs. not cells during culture

Reviewer #2: 1. Figure 1B is not explained clearly enough to understand the significance

2. Figure 3B and C: Is it possible that transfection of LCLs with multiple RNP complexes decreases the efficiency of knockout compared to one RNP which would give similar decreased results?

3. Figure 5A is presented as a percent of no inhibitor, but does Oleate have any effect on cell number when treated? Could these be shown as relative to no treatment at all to show the effects of palmitate and oleate on the untreated cells?

4. Panel 4G: Y-axis states relative cell number but looks to be actual cell number, not relative

Reviewer #3: Some minor comments should be addressed to enhance the manuscript.

1. It is appreciated that the authors choose their words carefully regarding dual SCD1 and FADS2 inhibitors. The work would be improved and conclusions strengthened if the authors formally tested if A939572 (SCD1i) and SC-26196 (FADS2i) are additive or synergistic (superadditive).

2. The study only examined some free fatty acids (FFA). Since they only looked at one, very narrow type of lipid, the authors should consider their usage of ‘lipidomics’. Since a full lipidomic analysis may yield additional or novel insight, it may be more appropriate to describe the current work as FFA analysis or FFA profiling.

3. Figs. 1, 3: The FADS2 blots are faint/hard to interpret. Could the authors provide better images? Relatedly, if the authors state the number of unique peptides that are contributing to the levels of SCD1 and FADS2 in Fig. 1D then the readers will be able to more appropriately judge the data.

4. Fig. 5A: It appears that 6 DMSO conditions were set to 100%. Why did the authors choose to do this instead of setting everything relative to DMSO 0µM palmitate / 0µM oleate?

5. Fig. 5D: Please remake this figure, it is impossible to read.

6. Fig. 5F: The desaturation reaction consumes oxygen. Does dual SCD1i + FADS2i treatment reduce the non-mitochondrial oxygen consumption rate?

7. Fig. 6A: Please consider changing the symbols or colors to enable the FADS2i data to be shown.

8. Editorial items:

-Line 89: are there only three human fatty acid desaturases? Please confirm.

-Lines 93-96: please provide the reference for this sentence.

-Line 432: Please define UFA as unsaturated fatty acid.

-Line 539: Did they mean palmitoleate instead of oleate?

-Reference 43 is a duplicate of reference 38. Same with reference 46/55. The other references should be checked.

PLOS authors have the option to publish the peer review history of their article (what does this mean? ). If published, this will include your full peer review and any attached files.

**Do you want your identity to be public for this peer review?** For information about this choice, including consent withdrawal, please see our Privacy Policy .

Reviewer #1: No

Reviewer #2: No

Reviewer #3: No

**Figure resubmission:**

**Reproducibility:**



---

## [Decision Letter · Decision Letter 1]

24 Apr 2025

Dear Dr. Luftig,

We are pleased to inform you that your manuscript 'Fatty acid desaturases link cell metabolism pathways to promote proliferation of Epstein-Barr virus-infected B cells' has been provisionally accepted for publication in PLOS Pathogens.

Best regards,

Benjamin E Gewurz, M.D., Ph.D.

Academic Editor

PLOS Pathogens

Alison McBride

Section Editor

PLOS Pathogens

Sumita Bhaduri-McIntosh

Editor-in-Chief

PLOS Pathogens

orcid.org/0000-0003-2946-9497

Michael Malim

Editor-in-Chief

PLOS Pathogens

orcid.org/0000-0002-7699-2064

Reviewer Comments (if any, and for reference):

Reviewer's Responses to Questions

**Part I - Summary**

Reviewer #2: The authors have adequately responded to this reviewers concerns.

Reviewer #3: This is a resubmission of a manuscript from Banglack and colleagues. The authors' responses were appropriate, and they made several changes to improve the manuscript. They addressed the previous critique by performing additional experiments and analyses and extensively rewrote the manuscript. The study will likely lead to future work that will further tease apart how the virus may be taking advantage of a host-driven process or help to drive the process. Nonetheless, this is a promising study, and the authors should be commended for their hard work.

**Part II – Major Issues: Key Experiments Required for Acceptance**

Reviewer #2: (No Response)

Reviewer #3: (No Response)

**Part III – Minor Issues: Editorial and Data Presentation Modifications**

Reviewer #2: (No Response)

Reviewer #3: (No Response)

PLOS authors have the option to publish the peer review history of their article (what does this mean? ). If published, this will include your full peer review and any attached files.

**Do you want your identity to be public for this peer review?** For information about this choice, including consent withdrawal, please see our Privacy Policy .

Reviewer #2: No

Reviewer #3: No

---

## [Editor Report · Acceptance letter]

Dear Dr. Luftig,

We are delighted to inform you that your manuscript, "Fatty acid desaturases link cell metabolism pathways to promote proliferation of Epstein-Barr virus-infected B cells," has been formally accepted for publication in PLOS Pathogens.

Best regards,

Sumita Bhaduri-McIntosh

Editor-in-Chief

PLOS Pathogens

orcid.org/0000-0003-2946-9497

Michael Malim

Editor-in-Chief

PLOS Pathogens

orcid.org/0000-0002-7699-2064